# Aliskiren Hemifumarate Proliposomes for Improved Oral Drug Delivery: Formulation Development, In Vitro and In Vivo Permeability Testing

**DOI:** 10.3390/molecules27154828

**Published:** 2022-07-28

**Authors:** Priyanka Kunamaneni, Surya Kovvasu, Steven Yeung, Jeffrey Wang, Salim Shah, Guru Betageri

**Affiliations:** 1Department of Pharmaceutical Sciences, College of Pharmacy, Western University of Health Sciences, Pomona, CA 91766, USA; pkunamaneni@westernu.edu (P.K.); skovvasu@westernu.edu (S.K.); skyeung@westernu.edu (S.Y.); jwang@westernu.edu (J.W.); 2Sarfez Pharmaceuticals, Inc., McLean, VA 22102, USA; salim.shah@sarfez.com

**Keywords:** aliskiren hemifumarate, proliposomes, PAMPA, Caco-2, pharmacokinetic studies

## Abstract

The objective of this study was to develop proliposomal formulations for a poorly bioavailable drug, aliskiren hemifumarate (AKH). A solvent evaporation method was used to prepare proliposomes using different lipids. The lipids of selection were soy phosphatidylcholine (SPC), dimyristoylphosphatidylcholine (DMPC), and dimyristoylphosphatidylglycerol sodium (DMPG Na), stearylamine, and cholesterol in various ratios. Proliposomes were evaluated for particle size, zeta potential, in vitro drug release, in vitro permeability, and in vivo pharmacokinetics upon hydration with aqueous phase. In vitro drug release studies were conducted in 0.01 N hydrochloric acid using USP type II dissolution apparatus. Parallel artificial membrane permeation assay (PAMPA) and Caco-2 cell line models were used to study the in vitro drug permeation. Male Sprague-Dawley (SD) rats were used to conduct in vivo pharmacokinetic studies. Among different formulations, proliposomes with drug/DMPC/cholesterol/stearylamine in the ratio of 1:5:0.025:0.050 (*w/w*/*w*/*w*) demonstrated the desired particle size, higher zeta potential, and higher encapsulation efficiency. The PAMPA and Caco-2 cell line experiments showed a significantly higher permeability of AKH with proliposomes as compared to pure AKH. In animal studies, the optimized formulation of proliposomes showed significant improvement in the rate and extent of absorption of AKH. Specifically, following a single oral administration, the relative bioavailability of AKH proliposome formulation was 230% when compared to pure AKH suspension.

## 1. Introduction

A major risk factor affecting most people worldwide is elevated blood pressure; an assessed 17.9 million people died from cardiovascular diseases in 2016 and that figure is projected to grow to more than 23.6 million by 2030 [1]. The treatment goal of hypertension is to lower blood pressure. As per the American Heart Association, blood pressure numbers of less than 120/80 mm Hg are considered within the normal range. Hypertension stage 1 is when blood pressure consistently ranges from 130/139 systolic or 80/89 mm Hg diastolic, and doctors are likely to recommend lifestyle changes and may consider prescribing blood pressure medication based on the risk of atherosclerotic cardiovascular disease (ASCVD), such as heart attack or stroke. Hypertension stage 2 is when blood pressure consistently ranges at 140/90 mm Hg or higher [2,3], and doctors are likely to prescribe a combination of blood pressure medications and lifestyle changes. The main pathway through which many of the pharmacological treatment options work is through renin-angiotensin-aldosterone system (RAAS) in which each category of drug intervenes the cycle at each stage [4].

The primary step for initiation of RAAS is production of renin from juxtaglomerular cells of the kidney [4]. Aliskiren hemifumarate (AKH) is a direct renin inhibitor, reducing plasma renin activity (PRA) and preventing the conversion of angiotensinogen to angiotensin I. Whether AKH affects other RAAS components, e.g., angiotensin converting enzyme (ACE) or non-ACE pathways, is not known [5].

The currently approved AKH indication is essential (primary) hypertension. Lowering high blood pressure helps to prevent nonfatal stroke, kidney complications, high blood potassium, and heart attacks. AKH is not recommended for children younger than 2 years due to increased risk of side effects [5,6]. AKH has the molecular formula C_64_H_110_N_6_O_16_ (Figure 1) and has a very low bioavailability (about 2.6%) [7], and as a result, the doses are high. The marketed available products have doses of 150 mg and 300 mg tablets, which are used for once daily dosing as AKH has long half-life of 23–36 h. AKH is also available in combination with hydrochlorothiazide and amlodipine. Based on the study carried out on AKH in human volunteers [8], the absorption of AKH was low when administered orally. The low absorption of AKH was due to poor permeability through intestinal membrane as AKH is a biopharmaceutical classification system (BCS) class III drug. Very few research studies have been conducted to improve the permeability of AKH. Nanoparticles were used to deliver AKH, but the authors only optimized the preparation process but did not study pharmacokinetic aspects [9].

In the present study we attempted to encapsulate AKH into phospholipid multilayer vesicles as proliposomes [10]. Most of the research studies used proliposomal formulations to improve the solubility of poorly soluble drugs [11,12,13,14]. Liposomes have two phases, therefore, they can act as carriers for both hydrophilic and hydrophobic drugs. One of the biggest challenges is encapsulating a hydrophilic drug into the proliposomes.

To improve the encapsulation of hydrophilic drugs, a few strategies were considered in order to improve zeta potential, particle size, and preparation methods [15]. The simplest approach is the use of surface charge modifiers which enhance the incorporation of an opposite charged drug into the aqueous compartment by electrostatic forces. The role of electrostatic interactions and surface charge in improving drug loading has been reported previously [16,17]. From the commercialized viewpoint, formulating AKH into proliposomes improves stability as they are a dry, free flowing powder which can be either compressed into tablets or can be filled into capsules. When these dosage forms encounter the biological fluids in the body, dispersions of liposomes are formed [18]. We believe this is the first work that shows a proliposomal formulation enhancing the oral bioavailability of AKH.

The main emphasis of this research was to formulate proliposomes using various phospholipids and charge modifiers to enhance the encapsulation of AKH. PAMPA and Caco-2 models were used to examine in vitro permeability, while a rat model was used to investigate in vivo bioavailability.

## 2. Results

The stability, handling, and storage issues associated with liposomes can be overcome by use of proliposomal technology. Proliposomes can be administered by oral, parenteral, topical, and other routes. Water-soluble and water-insoluble drugs are formulated as proliposomal formulations for oral drug delivery to enhance their oral bioavailability [16]. In the present study, AKH proliposomes were prepared and studied as a drug delivery system to enhance the permeability through the intestinal mucosa and decrease hepatic first pass metabolism, thereby increasing the oral bioavailability of AKH.

### 2.1. Analytical Method Validation

#### 2.1.1. Linearity and Range

The relation between AKH concentration and peak area was linear from 5–50 µg/mL (y = 15.031x − 3.3877, r^2^ = 0.999). In addition, blank samples that included mobile phase, dissolving solvent were also run to discard the presence of interferences.

#### 2.1.2. Precision and Accuracy

Percent RSD values for intra- and inter- run samples were less than 2% which was acceptable for all quality control samples. The accuracy of AKH in the quality control samples was detected in the range of 98–101%. Results shown in Table 1.

#### 2.1.3. Stability

The AKH solution’s stability was determined by calculating the % recovery for a period of 3 days at different conditions. The percent recovery values when stored at room temperature and refrigerated conditions were observed to be 98–102%. Stability results are shown in Table 2.

#### 2.1.4. Robustness

Samples were tested at flow rates 0.9, 1.0, 1.1 mL/min and the percent RSD values were less than 2%. The percent RSD values with varied mobile phase ratios of 50:40:10, 60:30:10, and 50:30:20 (*v*/*v*/*v*) were also less than 2% indicating the method to be very robust. Results shown in Table 3 and Table 4.

#### 2.1.5. System Suitability

Six replicate injections of the standard preparation were injected to determine asymmetry, number of theoretical plates, and relative standard deviation of the peak area. The resulted percent RSD value was less than 2.0%. Parameters shown in Table 5.

### 2.2. Formulation of Proliposomes

For the preparation of proliposomal formulations, lipids including SPC, DMPC, and DMPG Na were evaluated. Cholesterol was used to impart the bilayer rigidity of liposomes [19] and stearylamine was used as charge modifier. The composition of proliposomal formulations is summarized in Table 6. The free-flowing powder was filled into size “00” capsules.

### 2.3. Physical Characterization of Proliposomes

After hydrating the proliposomes with purified water, mixing was performed, and the hydrated liposomal formulations were examined under a microscope for morphology. Optical microscopy studies confirmed the formation of spherical structures and uniform distribution (Figure 2). The average particle size ranges were 1674/1989 nm for SPC formulations, 309/450 nm for DMPC formulations, and 596/647 nm for DMPG Na formulations. The particle size of DMPC formulations in the presence of stearylamine was 302/427 nm. The charge on the liposomes was neutral for all the proliposomal formulations in the absence of stearylamine. Proliposomes prepared with stearylamine resulted in a positive surface charge of 21.3–39.7 mV (Table 7).

### 2.4. Encapsulation Efficiency

Liposome lysis was achieved using absolute alcohol and sonicated for 10 min to determine the percentage of drug encapsulation. The encapsulation efficiencies were found to be 12%/23% in SPC and DMPG Na proliposomes. The encapsulation efficiency for DMPC proliposomes were 22% and 37% with 1:2 and 1:5 of drug/lipid ratio, respectively. Addition of stearylamine to DMPC proliposomes resulted in higher encapsulation (47%). This may be due to the electrostatic interaction between the drug and the liposomes favoring drug encapsulation (Table 7).

### 2.5. DSC Analysis

DSC studies of pure AKH showed a sharp melting point at 103.38 °C. The drug, when subjected to a cooling cycle followed by a heating cycle, showed a peak at 103.38 °C. The drug in the form of proliposomal powder formulation showed a peak at 55.58 °C (Figure 3). This change in the melting temperature of AKH in proliposome formulation is likely due to physical interaction between the lipid and the drug.

### 2.6. In Vitro Dissolution Studies

The in vitro dissolution profiles were obtained for the optimized formulations in 0.01 N HCl. The proliposomal formulations were optimized based on their particle size and entrapment efficiency. Among all the proliposomal formulations, compositions containing AKH, DMPC, cholesterol, and stearylamine at weight ratios 1:5:0.025:0.025, (F7) and 1:5:0.025:0.050, (F8), were selected for the dissolution studies because of their higher entrapment efficiency.

With microcrystalline cellulose (MCC) as a filling agent, prepared proliposomal formulation (F8) was filled into size “00” HPMC capsules. To determine the impact of MCC on dissolution profiles, formulas were evaluated in liposomal and proliposomal forms. (Dissolution profiles of proliposomes are discussed in Section 2.10). 

Further, liposomal formulations of F7 and F8 were prepared separately using a similar procedure as that which was followed for proliposomes, but without the addition of microcrystalline cellulose (MCC). Liposomal suspension was dried until there was a formation of thin film and it was free from ethanol. The prepared thin film was dispersed with nanopure water and tested for dissolution. The amount of dissolved AKH from pure drug was 75% in 10 min, 92% for F7, and 83% for F8 liposomal formulations (Figure 4). The statistical significance of AKH pure drug versus F7 (liposome), AKH pure drug vs F8 (liposome), and F7 (liposome) vs F8 (liposome) was determined using a 2-way ANOVA (Tukey’s multiple comparison test) in GraphPad Prism (viewing mode) 9.1.2 (226).

Since AKH is a highly soluble drug, dissolution is not the rate limiting factor, therefore the optimized F8 formulation ought to test for permeability studies.

### 2.7. Parallel Artificial Membrane Permeability Assay

The PAMPA model is often used to estimate the passive transcellular permeability due to lack of transporter and pore mediated permeability. An acceptable lipophilicity (log P) is necessary for a drug to travel across the phospholipid membranes by passive diffusion. 

Permeability of pure AKH and proliposomal formulation in the PAMPA model are given in Figure 5. Up to a maximum concentration of 5% DMSO is used to dissolve the drug compound which was sufficient for the proliposomal formulations and pure AKH. The flux (J) values for the pure AKH and DMPC proliposomes F8 were found to be 0.554 ± 0.003 and 0.847 ± 0.008, respectively (Table 8). These results indicate that DMPC proliposomal formulations have shown enhanced permeability compared to pure AKH. About 1.5-fold increase in flux was observed with DMPC formulation F8 compared to pure AKH. Statistical analysis of the permeability coefficient of DMPC formulations and pure AKH powder (*p* < 0.05) proved that there was a significant increase in the permeability of AKH when formulated into proliposomes. Since the PAMPA study showed that DMPC formulations result in higher permeation compared to pure AKH, further studies that include Caco-2, DSC analysis, stability studies, and pharmacokinetic studies were performed on the DMPC formulation.

### 2.8. Caco-2 Studies

Transport studies were performed with Caco-2 monolayers which were 21–25 days old and grown on filter inserts. Initial integrity of the monolayers was tested, and the TEER value was 333 ± 25 Ω cm^2^ at 37 °C. At the end of permeation studies, the TEER value was tested and observed to be 319 ± 36 Ω cm^2^. The TEER values imply that the integrity of the Caco-2 monolayer was maintained throughout the study. An 8.057-fold increase in apparent permeability (apical to basal) was observed with proliposomal formulation F8 when compared with AKH alone (Table 9). TEER values suggest that higher permeation was not due to modulation of the tight junctions and or a compromised membrane, but possibly due to transcellular pathways.

### 2.9. In Vivo Studies

AKH extracted from the plasma using liquid-liquid extraction showed well resolved chromatographic peaks. The blank plasma after extraction contained no significant interference peaks. Hydrochlorothiazide was used as internal standard (IS). The relation between AKH concentration and peak area ratios of AKH to hydrochlorothiazide was linear from 20–100 ng/mL (y = 487.05x − 14,371.3, r^2^ = 0.9857). LCMS/MS chromatograms of hydrochlorothiazide (IS) and AKH are shown in Figure 6. Precision and accuracy were determined based on the calibration curve. Selected pharmacokinetic parameters of pure AKH and liposomal formulations are shown in Figure 7 and Table 10. The Tmax value of pure AKH and proliposomal formulation (F8) were found to be 0.5 h. Cmax of pure AKH was 49.12 ng/mL that of F8 formulation was 116.21 ng/mL. The AUC_0–12_ h of proliposomal formulations was significantly more than that of AKH that resulted in a relative bioavailability of proliposomal formulation of 230%. Statistical significance of AUC_0–12_ h was determined using an unpaired Student’s t-test (two tailed, 95% C.I.) keeping the plain drug data as control. Proliposomal formulation (F8) had shown a significant increase over plain drug (*p* < 0.0001).

### 2.10. Stability Studies

The capsules with F8 formulations were filled into a 150 mL HDPE bottle and used for a stability study. Weight and lock length was tested as a part of physical evaluation of capsules on day zero. The weight of the proliposomal capsules was found to be 298–315 mg. Percent drug release from F8 (proliposome) demonstrated up to a 10% difference between the initial and 3 months accelerated temperature (40 °C/75 percent RH) until 45 min (Figure 8). The statistical significance of the AKH pure drug versus F8 (proliposome) initial, and F8 (proliposomes) initial vs F8 (proliposomes) at stability is determined using a 2-way ANOVA (Tukey’s multiple comparison test), GraphPad Prism (viewing mode) 9.1.2 (226).

Similarity factor *f2* was used to better compare the dissolution profiles. Proliposomal formulation (F8) at initial and 3 months stability showed a similarity factor (*f2*) 57.46. The particle size and zeta potential for 3 months stability samples were found to be 367.8 ± 43 nm and +35.2 mV, respectively.

## 3. Discussion

Liposomes are unilamellar or multilamellar spheroid structures composed of lipid molecules, often phospholipids. On oxidation, the solubility of liposomes is increased, and they have a tendency to aggregate or fuse to hydrolysis. Proliposomes are used as another possibility for liposomes which are composed of water-soluble porous powder as carrier phospholipids and drugs dissolved in organic solvent. Drug and phospholipid materials are coated on carrier material to form free flowing granular material which demonstrates better stability, solubility, and controlled release [20]. In the present study, proliposomal formulations for improving the oral bioavailability of AKH were attempted. AKH is a novel antihypertensive agent. The absolute bioavailability of AKH is only 2.6%. The lower bioavailability is due to poor permeation. Proliposomal formulations were formulated to increase the bioavailability. The developed HPLC method was optimized, validated for various experimental parameters, and provides high throughput for the determination of AKH with excellent linearity, accuracy, precision, stability, robustness, and system suitability. Proliposomal formulations were successfully prepared using various compositions of lipid, cholesterol, MCC, and charge modifiers. Cholesterol was included in the formulations to prevent the loss of hydrophilic materials and stabilize lipid bilayer by decreasing the rotational freedom of phospholipid hydrocarbon chains [21]. Proliposomes were hydrated in purified water and vortex mixed to obtain liposomes and further tested for particle size and surface charge. The particle size of all formulations ranged from 427 nm-1.99 µm. Vesicle size is a useful parameter which aids in determining physical stability such as aggregation and fusion of liposomes. Aggregation triggers a shift in mean size and size distribution of liposomes toward higher values and causes destabilization of liposomes [22]. The particle size of DMPC formulations (F3 and F4) were lower compared to other lipid formulations. It postulates if the particle size is small the transport across the membrane is better. The charge of all liposomes was found to be neutral. Among SPC, DMPC, and DMPG Na proliposomes, DMPC formulations had the highest entrapment efficiency, encapsulating 37% of AKH with a drug/lipid cholesterol ratio of 1:5:0.025. The higher encapsulation with DMPC proliposomes might be due to saturated phosphatidylcholine which forms a rigid, rather impermeable bilayer, whereas unsaturated lipids from natural sources (i.e., egg or soybean phosphatidylcholine) contribute more permeable and less stable bilayers [23]. The lower encapsulation of DMPG Na formulations could also be due to a weak bilayer structure and decreased packaging space for accommodating the drug molecule which might have resulted in drug leaking [24]. When stearylamine was incorporated (F7, F8), it resulted in positive surface charge. Positive charge on the liposomes will prevent aggregation, further, liposomes with positive charge will be prone to binding with negatively charged cell membranes due to electrostatic interaction [25]. The role of surface charges on proliposomal formulations was reported by Janga et al. [26] and Velpula et al. [27]. The inclusion of charge modifiers (stearylamine) resulted in higher encapsulation (47%) with the F8 formulation and was considered as the optimized formulation. Increased EE may be explained by the electrostatic interactions between the positively charged phospholipid bilayer and the aliphatic chain attached to the cyclic core of drug favoring drug encapsulation. Stearylamine is a cationic lipid, which provided a positive charge on surface of hydrated proliposomes. Increase in the encapsulation could be due to the electrostatic attractions between the positive charge of phospholipid and aliphatic chain of the aliskiren drug.

When the dissolving profiles of F8 formulations as liposomes and proliposomal formulations (MCC as filling agent) were compared, proliposomal powder formulation release was observed to be delayed and slower than liposomal formulations, possibly due to greater lipid interactions [28]. 

Morphological evaluation of liposomes formed after hydration of proliposomes with purified water revealed the formation of spherical structures when observed under an optical microscope. Proliposomal capsules (F8) were tested for dissolution studies and cumulative percent drug release was found to be 61% and 88% at 60- and 120-min time intervals, respectively. Cromolyn, a BCS class III drug, demonstrated better permeability with lipid formulations than with a pure drug in prior tests conducted in our lab [29]. As AKH is a water-soluble drug, dissolution of the drug is not the rate limiting step. Permeability studies using PAMPA showed a 1.5-fold increase in flux for F8 formulation when compared to pure AKH. A Caco-2 study showed an 8-fold increase in apparent permeability, and a higher uptake of the drug in Caco-2 monolayers might be due to the internalization of AKH loaded proliposomes.

Proliposomal powder formulation (F8) was filled into size “00” capsules with MCC as a filling agent. The stability results of F8 proliposomal formulation indicate a drop in cumulative percent drug release in the initial time points at 3 months, 40 °C/75% RH condition which might be due to a sticky solid mass of powder. However, the release profiles are comparable in the later time points. Based on 3 months accelerated stability data, 25 °C/60% RH and refrigerated (2–8 °C) conditions are preferred for product storage.

Proliposomal formulation resulted in increased absorption with no change in the rate of absorption. Consequently, Cmax and AUC increased but Tmax remained the same. Pharmacokinetic studies in rats revealed that plasma concentration of proliposomal encapsulated AKH was significantly higher than that from pure AKH solution demonstrating that proliposomes were successful in increasing AKH bioavailability. 

## 4. Materials and Methods

### 4.1. Materials

Aliskiren hemifumarate was obtained from Nantong Chanyoo Pharmatech Co., Ltd., (Nantong, China). Dimyristoylphosphatidylglycerol sodium (DMPG Na) was acquired from Gen-zyme Pharmaceuticals (Cambridge, MA, USA). 1,2-Dimyristoyl-*SN*-glycero-3-phospho-choline (DMPC) (COATSOME MC- 4040 EX) was procured from NOF Corporation (Ebisu Shibuya-Ku, Japan). Soy Phosphatidylcholine (SPC) was acquired from Avanti Polar Lipids (Alabaster, AL, USA). Cholesterol, stearylamine, and magnesium stearate were obtained from Spectrum Chemical and Laboratory Products (Los Angeles, CA, USA). Avicel PH 102 was obtained from FMC BioPolymer (Philadelphia, PA, USA). Parallel artificial membrane permeability assay (PAMPA) plates were obtained from Millipore (Billerica, MA, USA). Caco-2 plates (6-well) were procured from Corning Incorporated Costar (Kennebunk, ME, USA), Dulbecco’s Modified Eagle’s Medium (DMEM) with 4.5 g/L glucose and without L-gluta-mine and sodium pyruvate, Hank’s Balanced Salt Solution with calcium and magnesium and without phenol red (HBSS), penicillin 10,000 U/mL streptomycin 10,000 mg/mL solution (100×), Dulbecco’s phosphate-buffered saline without calcium and magnesium (DPBS), and trypsin/EDTA solution were obtained from Corning™ Cellgro™ (Kennebunk, ME, USA). Polycarbonate membrane (pore size 0.2 mm, diameter 6 mm, was obtained from Corning (USA). Pre-cannulated Sprague-Dawley rats were obtained from Envigo (San Diego, CA, USA). Sprague-Dawley rat plasma was obtained from Innovative Research, Inc. (Court Novi, MI, USA). All other materials and organic solvents were of HPLC grade and were obtained from EMD (Billerica, MA, USA).

### 4.2. Analysis of Aliskiren Hemifumarate

AKH analysis for encapsulation efficiency, drug content, in vitro drug release, and in vitro permeation studies were carried out using a reversed phase HPLC method. Agilent 1100 HPLC system (UV-visible detector) was used for the analysis. The mobile phase consisted of 50 mM tris buffer pH adjusted to 3.0 using orthophosphoric acid, acetonitrile, and methanol (50:40:10 *v*/*v*/*v*). Flow rate was set at 1.0 mL/min. The injection volume was 20 µL. The chromatographic separation was achieved on Phenomenex C-18 column (5 µ m, 250 × 4.6 mm) and the UV detector was set at 230 nm.

#### Preparation of Calibration Standards and Quality Control Samples

10 mg of AKH was dissolved in 10 mL of water. The stock solution was further di-luted with the dissolving solvent to obtain different working standard solutions ranging from 5–50 µg/mL. Quality control samples for AKH were prepared at low (5 µg/mL), medium (20 µg/mL) and high (50 µg/mL) concentrations. All the solutions were prepared in triplicates obtained from three different stock solutions.

### 4.3. Method Validation

#### 4.3.1. Linearity and Range

Linearity was evaluated using freshly prepared standard solutions. Linearity was covered from the range of 5–50 µg/mL of AKH. In addition, blank samples that include mobile phase, dissolving solvent were also run to discard the presence of interferences.

#### 4.3.2. Precision and Accuracy

For determination of intra-day precision, inter-day precision, and accuracy, quality control samples were analyzed both within a single instrument run and in different runs. The accuracy was calculated from the ratio of measured concentration, based on the standard curve, to the nominal added concentration. Precision was evaluated by calculating the inter-day and intra-day variation i.e., relative standard deviation of measured concentration (%RSD) for a period of 3 days.

#### 4.3.3. Stability

To determine stability, quality control samples had been stored at room temperature and at refrigerated temperature. The samples used for inter- and intra-day precision were also used to determine the stability of the samples at room temperature. In order to determine the stability at refrigerated conditions, samples were withdrawn from the fridge each day, until the third day, and analyzed. The stability was determined by calculating the %RSD for a period of 3 days.

#### 4.3.4. Robustness

Flow rate, mobile phase composition, and the column temperature were altered and checked for the effect of these changes on the signal response to determine the robustness of the method. Three different flow rates 0.9, 1.0, and 1.1 mL/min, three different mobile phase compositions that include 50:40:10, 60:30:10, and 50:30:20 *v*/*v*/*v* mixture of 50 mM triethylamine buffer pH 3.0, acetonitrile, and methanol were evaluated to determine the robustness of the method. %RSD was calculated for individual parameters to determine the robustness.

#### 4.3.5. System Suitability

The chromatographic system was tested for suitability prior to each stage of validation. Six replicate injections of the standard preparation were injected, and the asymmetry number of theoretical plates and relative standard deviation of the peak area were determined.

### 4.4. Preparation of Proliposomes

Formulations of AKH were prepared with SPC, DMPC, DMPG Na, cholesterol, and surface charge modifier stearylamine. Proliposomes were prepared by solvent evaporation method [30]. During the process, lipid and cholesterol were dissolved in ethanol. Following the addition of the drug, the resultant dispersion was adsorbed onto microcrystalline cellulose (MCC). Drug dispersion equivalent to 100 mg of aliskiren was absorbed onto 300 mg of #60 mesh (A.S.T.M. E-11 U.S. Standard Sieve Series, 250 µL, Dual Manufacturing Co., Franklin Park, IL, USA) passed MCC. The dry powder was laid open to vacuum desiccation to remove any solvent residues and passed through the #60 mesh sieve to achieve free flowing proliposomal powder. 

### 4.5. Characterization of Formulations

#### 4.5.1. Particle Size

Proliposomes were hydrated in purified water and vortex mixed for a minute to obtain liposomes. The resultant suspension was then analyzed for size distribution using a Malvern Zetasizer ZS90 (Malvern Instruments, Grovewood Rd, Malvern WR14 1XZ, UK).

#### 4.5.2. Morphological Evaluation

The formation and morphology of proliposomes were evaluated using optical microscopy. For morphological evaluation, proliposomal powder was dispersed in purified water and mixed gently. The liposomes formed after hydration was observed through confocal microscope (Leica, Microsystems Italia, Milan, Italy). Magnification of 40× was used to capture the image.

#### 4.5.3. Encapsulation Efficiency

The percentage of drug encapsulated was determined after lysis of the prepared liposomes with absolute alcohol and sonication for 10 min. The concentration of AKH in absolute alcohol was determined spectrophotometrically at 230 nm using HPLC. The encapsulation efficiency was calculated using the following formula [28,31]:Encapsulation efficiency (%) = (Total drug − Free drug)/Total drug × 100

#### 4.5.4. Differential Scanning Calorimetric (DSC) Analysis

To evaluate the interaction between the drug and other excipients used in the formulation, differential scanning calorimetric (DSC) studies were conducted. DSC studies of pure AKH, and different combinations of AKH, along with the excipients used in the formulations were performed using a DSC-Q2000 coupled to a refrigeration unit RC-40 (TA Instruments, USA). Samples between 3 and 8 mg were weighed into a flat bottom aluminum pan and hermetically sealed using a manual press. A pre-weighed empty equivalent pan was used as reference. After carefully placing both pans in the DSC cells, the furnace lid was closed. The experiment setup was a heat-cool-heat cycle with a scanning temperature starting at 40 °C to 200 °C at a heating rate of 10 °C/min and cooling rate of 10 °C/min and back to 40 °C.

### 4.6. Preparation of Capsule Dosage Form

The proliposomal formulations were sieved through #60 mesh (USA Standard Sieve Series, American Standard Test Sieve, Dual Manufacturing Co., Franklin Park, IL, USA). The proliposomal powder equivalent to 50 mg of aliskiren containing different compositions was filled into size “00” HPMC capsules (Capsugel, Greenwood, SC, USA) manually.

### 4.7. In Vitro Dissolution Studies

Dissolution testing for AKH was performed using 500 mL of 0.01 N HCl with a paddle at 50 rpm as per FDA recommendation. Two capsules, each equivalent to 50 mg of aliskiren were dropped into a dissolution beaker for testing the dissolution profile. In the case of liposomes, liposomal suspension equivalent to 100 mg was transferred to dissolution beaker. Sampling aliquots of 5.0 mL were withdrawn at 10, 20, 30, 45, and 60 min. No volume replacement was performed for the withdrawn and corresponding volumes at each time point were used in the calculation. After the end of each test time, sample aliquots were filtered through a 0.45 μm membrane nylon filter, and then analyzed by the HPLC method described in the previous section. Cumulative percent of drug release from the capsules was calculated and the mean of the three samples (*n* = 3) was used in the data analysis.

### 4.8. Parallel Artificial Membrane Permeability Assay (PAMPA)

PAMPA studies for AKH formulations were carried out using a published protocol [32]. A 96-well filter plate and 96-well receiver plate were used as the permeation donor and permeation receptor, respectively. Phosphate buffer, pH 6.8, was used as receptor and donor buffer. 300 µL of the pH 6.8 phosphate buffer was added to each well of the receptor/acceptor plate (made of polytetrafluoroethylene (PTFE)). 5 µL of previously prepared clear solution of 1% lecithin in dodecane (*w*/*v*) solution was carefully transferred to the wells of the donor plate. 1% lecithin solution was used as an artificial membrane and applied to the polyvinylidene difluoride (PVDF) membrane filter in the filter plate (donor plate). Immediately after the application of the artificial membrane, 150 µL of the proliposomal suspension and pure drug solution was added into each well of the filter plate. Then the drug-filled filter plate was carefully placed upon a receptor plate, ensuring liquid contact between the buffer in the receptor plate and the PVDF membrane filter. The whole donor receptor assembly was covered with a lid plate to avoid evaporation and contamination. The whole donor receptor assembly was incubated at room temperature, samples from the donor and the receptor component were withdrawn (*n* = 3) at scheduled time intervals and assayed for drug concentration using HPLC. The flux J (μg/cm^2^/h) was calculated following an earlier reported formula [33,34]. Where (dc) is the steady state of the mass transport, (dt) is unit time, and (A) is the area.

### 4.9. Caco-2 Permeation Studies

Caco-2 studies for AKH formulations were carried out using the following protocol [35,36]. Permeability studies were performed using a cultivated monolayer of Caco-2 cells in 75 cm^2^ plastic culture flasks. As culture media, DMEM with 1% (*v*/*v*) non-essential amino acid solution, 10% (*v*/*v*) fetal bovine serum (FBS), Na-pyruvate, and penicillin-streptomycin was used. The cells were kept at 37 °C with 5% CO_2_ and 90% relative humidity. The passage of cells was performed every 5 days at 1:4 dilution (at 70–80% confluence) using trypsin-0.025% EDTA. Passages 43–48 were used to study the transport and uptake. For this study, cells with >80% confluent were harvested with trypsin and 0.025% EDTA and seeded at a density of 2.5 × 10^5^ onto polystyrene inserts (24 mm diameter, 3.0 µm pore size, Costar, Cambridge, MA, USA) within the 6-well culture plates. After a resting period of 5 days after seeding, the culture medium from both the apical (1.5 mL) and the basolateral side (2.5 mL) was replaced every other day thereafter. After 22 days, Caco-2 cells formed a monolayer through which the transport studies were conducted. In preparation for the study, the monolayers were washed with HBSS and readings of the transepithelial electrical resistance (TEER) were recorded using an EVOM voltmeter connected to chopstick electrodes (World Precision Instruments, Sarasota, FL, USA). The washing media was removed by aspiration from both sides of the monolayer. Transport studies were performed by mixing either the sample formulations or the pure drug with 1.5 mL of HBSS (pH 7.4) medium at a final concentration of 100 µg/mL. This mixture was added to the apical (donor) side while fresh HBSS was added on the basolateral (receiver) side of the monolayer. After predefined intervals, 500 µL aliquots were removed from the receiver side and concentration of the drug was determined by HPLC. The study was performed in triplicates and the mean value was used in the data analysis. The apparent permeability coefficient (apical to basal) was calculated as follows:$$\mathrm{Apparent}\;\mathrm{permeability}\;(\mathrm{Papp})=\frac{V/\left(\mathrm{area}\;\times \;\mathrm{time}\right)\times \left[\mathrm{drug}\right]\;\mathrm{receptor}\;\mathrm{chamber}}{\left[\mathrm{drug}\right]\;\mathrm{initial},\;\mathrm{donor}\;\mathrm{chamber}}$$
where V is the volume of the receptor chamber in mL (2.5 mL), [drug] receptor chamber is the concentration of the drug in the basolateral side, [drug] initial, donor chamber is the concentration of the drug added to the apical side, area is the area of membrane insert (4.7 cm^2^) and time is the total exposure time in seconds.

### 4.10. Stability Studies

Stability studies were performed on AKH proliposomal formulations by exposing them to accelerated storage conditions (40 °C/75% RH) for 3 months. Capsules prepared were filled into HDPE bottles with 1.5 g of cotton. Capsules were removed at predetermined times at 30, 60, and 90 days and evaluated for physical appearance, drug content, and dissolution profiles. Content and dissolution analysis were carried out using the HPLC method developed for AKH formulations.

### 4.11. Pharmacokinetic Study

Animal study protocol was approved by IACUC (Western University of Health Sciences, Pomona). Healthy male Sprague-Dawley rats (225–250 g) with a jugular vein cannulation were used in the study. The animals were divided into 2 groups (*n* = 6) and then fasted overnight. Food was provided 2 h after administration of the drug. Group 1 was given a suspension of pure AKH, and Group 2 was given a suspension of AKH proliposomes; both groups were dosed by oral gavage at 7.5 mg/kg. Both pure AKH and the proliposomal suspension were suspended in 5% HPMC solution. 0.35 mL of blood samples were collected from the jugular vein catheter at 0 (before dosing), 0.5, 1, 2, 3, 4, 6, 8, 12, and 24 h. Blood samples were collected into pre heparinized Eppendorf tubes, centrifuged at 3000 rpm at 4 °C for 30 min, and plasma was collected and stored at −80 °C until analysis. Animals were sacrificed using a lethal dose of isoflurane.

### 4.12. Quantification of Aliskiren in Plasma Using LC/MS/MS

A stock solution (1 mg/mL) of AKH and hydrochlorothiazide (internal standard) was prepared in acetonitrile: water (90:10 *v*/*v*). Working solutions of aliskiren (20–100 ng/mL) were made in the same solvent. A liquid-liquid extraction method was employed for the extraction of drug from the plasma samples. Plasma samples (100 µL) were aliquoted into Eppendorf tubes and the working solutions of the internal standard were spiked into the plasma and mixed well by vortex. 400 µL of ethyl acetate was added to the mixture and vortexed vigorously. This is followed by centrifugation at 10,000 rpm for 10 min at 4 °C. The organic layer (top) was transferred into a new clean dry glass tube and evaporated to dryness by a steam of nitrogen at room temperature. To the residue, 100 µL of 50% methanol was added for reconstitution. 5 µL of sample was injected to LC/MS/MS system for analysis of the drug.

The chromatographic separation was achieved using a Zorbax SB-C18 column (Agilent, Santa Clara, CA, USA; particle size 5 µm; 2.1 × 150 mm) with the corresponding guard column (12.5 × 2.1 mm). An isocratic mobile phase, consisting of a mixture with ammonium acetate (2 mM) in 0.1% formic acid and acetonitrile (15:85 *v*/*v*) was used with the flow rate of 0.30 mL/min. The sample column was maintained at room temperature. A triple-quadrupole mass spectrometer (API 3200, Framingham, MA, USA) was used as detector. Samples were quantified using multiple reaction monitoring (MRM) mode with the transition of parent-to-daughter ions of 550.279 → 114.941 m/z for aliskiren and 296.900 → 77.846 m/z for hydrochlorothiazide. The operating conditions were Curtain Gas (CUR), 10; Collision Gas (CAD), 5; Ion Spray Voltage (IS), 4500; Temperature (TEM), 550; Ion Source Gas 1 (GS1), 40; Ion Source Gas 2 (GS2), 40.

## 5. Conclusions

The results show that using proliposomes as a carrier increased the bioavailability of aliskiren. Aliskiren belongs to BCS class III, which has a limited oral bioavailability due to its high solubility and poor permeability. Aliskiren’s therapeutic usage was aided by proliposomes, which improved the drug’s permeability and pharmacokinetic qualities. Among the various formulas prepared, the proliposomal formulation prepared with DMPC and stearylamine provided a positive charge of +39.7 mV which resulted in a higher encapsulation efficiency of 47.2% with a mean particle size of 302 nm. PAMPA and Caco-2 in vitro permeability experiments demonstrated a considerable increase in drug penetration. In vivo pharmacokinetic studies showed that employing proliposomes as the drug delivery system increased AKH bioavailability. The combination effect of greater permeability and increased lymphatic absorption could explain increased bioavailability. According to these findings, proliposomes could be employed as carriers for effective oral drug administration of water-soluble drugs like AHK. More research is needed to demonstrate improved bioavailability in human subjects and gain a better knowledge of lymphatic transport.

## Figures and Tables

**Figure 1 molecules-27-04828-f001:**
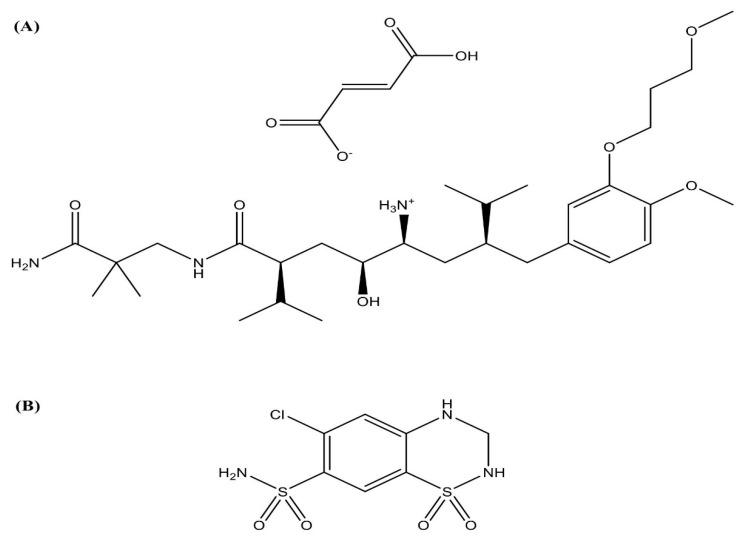
Chemical structures of (**A**) aliskiren hemifumarate and (**B**) hydrochlorothiazide (IS).

**Figure 2 molecules-27-04828-f002:**
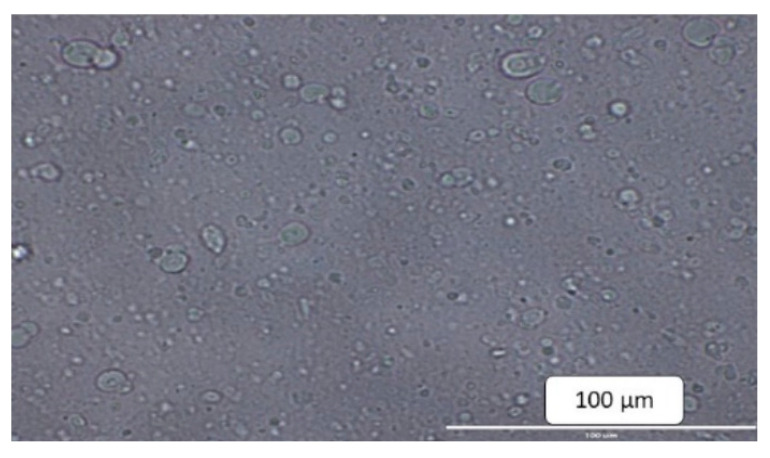
Microscopic image of proliposomes (F8) after hydrating with purified water.

**Figure 3 molecules-27-04828-f003:**
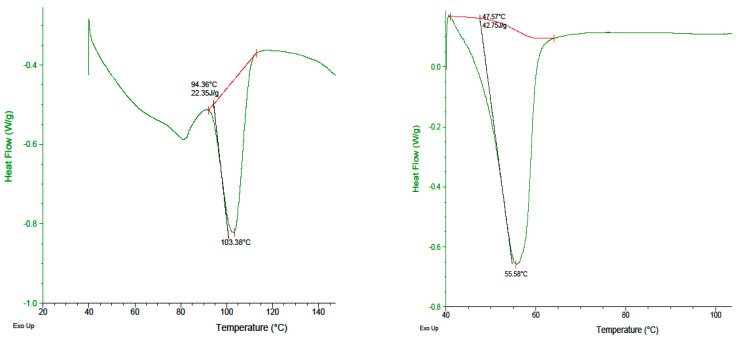
DSC thermograms of AKH and proliposomal formulation (F8).

**Figure 4 molecules-27-04828-f004:**
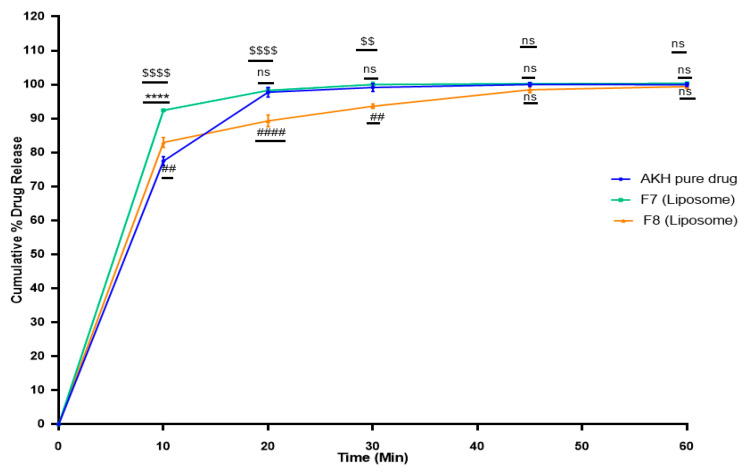
In vitro release profile of AKH pure drug and liposomal formulations in 0.01 N HCl (*n* = 3). Time (min) vs. Cumulative % drug release; $$$$—F7 (Liposome) vs. F8 (Liposome) with significance *p* < 0.0001; $$—F7 (Liposome) vs. F8 (Liposome) with significance *p* = 0.0012; ****—AKH Pure drug vs. F7 (Liposome) with significance *p* < 0.0001; ####—AKH Pure drug vs. F8 (Liposome) with significance *p* < 0.0001; ##—AKH Pure drug vs. F8 (Liposome) with significance *p* = 0.0086; ns—F7 (Liposome) vs. F8 (Liposome), AKH Pure drug vs. F7 (Liposome), AKH Pure drug vs. F8 (Liposome) are non-significant.

**Figure 5 molecules-27-04828-f005:**
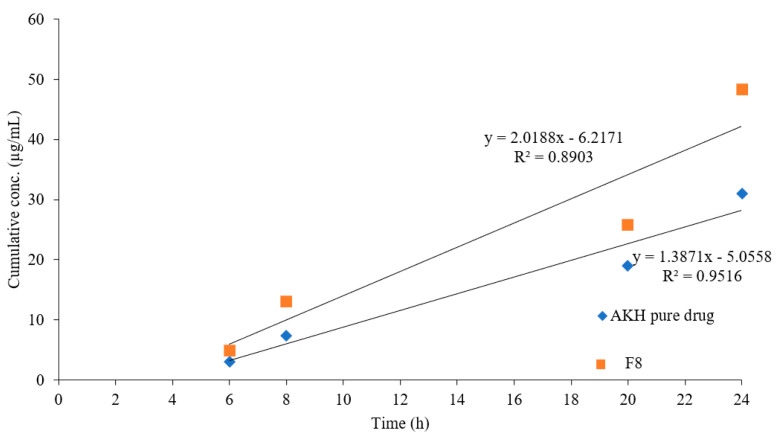
Average cumulative concentrations of AKH and proliposomal formulation.

**Figure 6 molecules-27-04828-f006:**
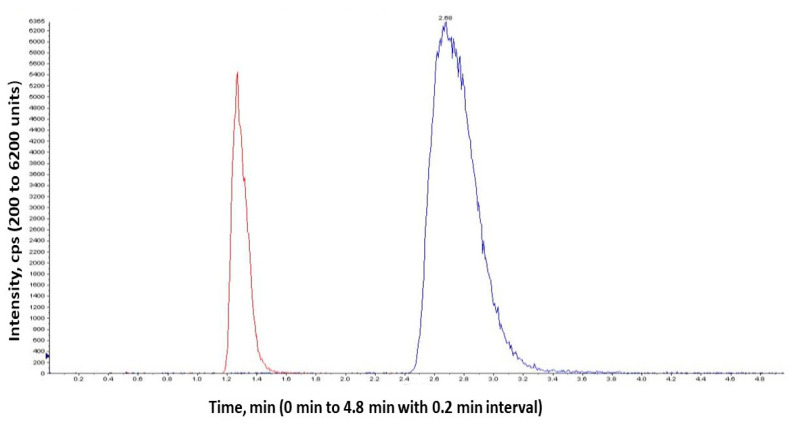
LCMS/MS chromatograms of hydrochlorothiazide (IS) and aliskiren hemifumarate.

**Figure 7 molecules-27-04828-f007:**
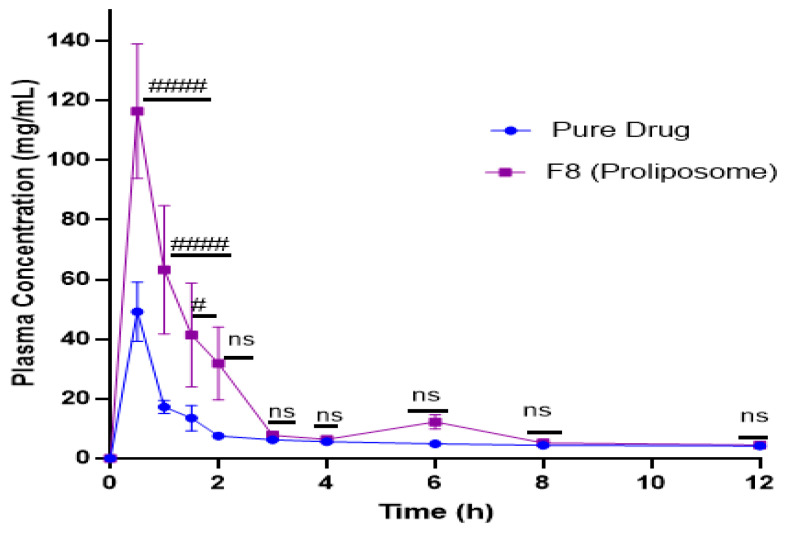
Mean plasma concentrations of AKH following oral administration of pure drug and its proliposomal formulation (F8). ####—Pure drug vs. F8 (Proliposome) with significance *p* < 0.0001; #—Pure drug vs. F8 (Proliposome) with significance *p* = 0.0409; ns—non-significant.

**Figure 8 molecules-27-04828-f008:**
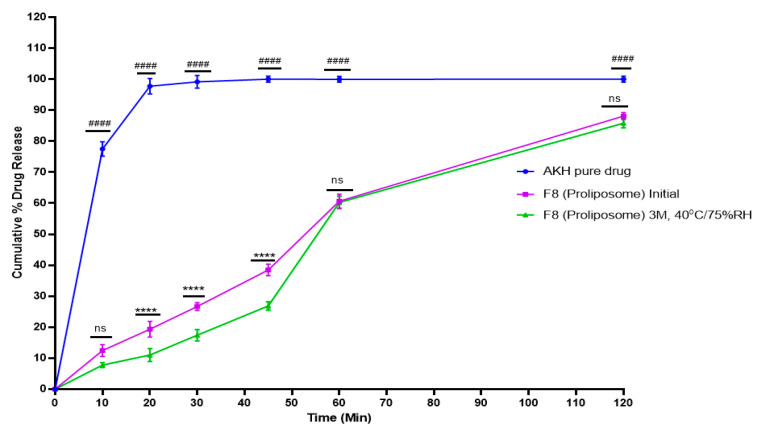
In vitro release profile of AKH pure drug and proliposomal formulation of initial and 3 M stability samples in 0.01 N HCl (*n* = 3). Time (min) vs. Cumulative % drug release; ####—AKH pure drug vs. F8 (Proliposome) Initial with significance *p* < 0.0001; ****—F8 (Proliposome) Initial vs. F8 (Proliposome) 3 M, 40 °C/75% RH with significance *p* < 0.0001; ns—F8 (Proliposome) Initial vs. F8 (Proliposome) 3 M, 40 °C/75% RH is non-significant.

**Table 1 molecules-27-04828-t001:** Intra-day, inter-day, and % recovery studies.

Drug Conc. (µg/mL)	Intra-Day Precision (*n* = 3) (%RSD)	Inter-Day Precision (*n* = 3) (%RSD)	% Recovery (*n* = 3)
5	0.38	0.36	98.84
20	1.65	1.15	100.39
50	0.32	0.95	98.88

**Table 2 molecules-27-04828-t002:** Percent recovery studies for stability samples kept at room temperature (RT) and refrigerator (2–8 °C).

Drug Conc. (µg/mL)	% Recovery Samples Kept at RT			% Recovery Samples Kept at 2–8 °C		
	(Day 1)	(Day 2)	(Day 3)	(Day 1)	(Day 2)	(Day 3)
5	98.08	98.97	99.24	99.17	97.56	100.59
20	100.06	99.95	99.43	100.41	101.05	101.74
50	101.35	99.49	99.33	99.72	100.61	98.67

**Table 3 molecules-27-04828-t003:** Robustness with varied flow rates (AKH Conc: 100 µg/mL).

Flow Rate (mL/min)	Retention Time (RT)	Area	%RSD
0.9	3.96	1256.46	0.25
1.0	4.2	1452.40	0.28
1.1	4.7	1536.53	0.28

**Table 4 molecules-27-04828-t004:** Robustness with varied mobile phase ratios (AKH Conc: 100 µg/mL).

Mobile Phase (*v*/*v*/*v*) Retention Time (RT)		Area	%RSD
50:40:10	4.2	1452.4	0.28
60:30:10	3.9	1493.6	0.24
50:30:20	4.6	1596.2	0.30

**Table 5 molecules-27-04828-t005:** System suitability parameters.

S.No.	Concentration of Std Taken	Area
1		1378.4
2		1384.0
3	100 µg/mL	1387.1
4		1379.6
5		1372.9
6		1375.7
Mean	1379.61	
%RSD	0.37	
Theoretical plate	20152	
Retention time	4.2	

**Table 6 molecules-27-04828-t006:** Formulation composition of proliposomes.

Formulations	Lipid	Drug (in Parts)	Lipid (in Parts)	Cholesterol (in Parts)	Stearylamine (in Parts)
F1	SPC	1	2	0.010	-
F2	SPC	1	5	0.025	-
F3	DMPC	1	2	0.010	-
F4	DMPC	1	5	0.025	-
F5	DMPG Na	1	2	0.010	-
F6	DMPG Na	1	5	0.025	-
F7	DMPC	1	5	0.025	0.025
F8	DMPC	1	5	0.050	0.050

**Table 7 molecules-27-04828-t007:** Particle size, zeta potential, and encapsulation efficiency for proliposomal formulations (*n* = 3).

Formulations	D:L:C (in Parts)	Stearylamine (in Parts)	Surface Charge (mV)	Entrapment Efficiency (%)	Particle Size (Mean ± SD, nm)
F1	1:2:0.010	-	0.034	17.5	1989 ± 87
F2	1:5:0.025	-	0.236	23.4	1674 ± 58
F3	1:2:0.010	-	0.019	22.0	309 ± 34
F4	1:5:0.025	-	0.543	37.0	450 ± 22
F5	1:2:0.010	-	0.923	12.1	647 ± 26
F6	1:5:0.025	-	0.236	14.1	596 ± 120
F7	1:5:0.025	0.025	+21.3	40.3	427 ± 35
F8	1:5:0.025	0.050	+39.7	47.2	302 ± 24

**Table 8 molecules-27-04828-t008:** Average flux (J) of AKH and proliposomal formulation (*n* = 3).

Formulation	Flux (J) (±SD)	No of Folds
		Increase in Flux
Pure AKH	0.554 (±0.003)	-
Proliposome (F8)	0.829 (±0.083)	1.497

**Table 9 molecules-27-04828-t009:** Apparent permeability of AKH and proliposomal formulation (*n* = 3).

Formulation	Media	Papp (cm s^−1^) (±SD)
Pure AKH	HBSS (Normal)	1.778 × 10^−6^ (±0.23)
Proliposome (F8)	HBSS (Normal)	1.433 × 10^−5^(±0.14)

**Table 10 molecules-27-04828-t010:** Selected pharmacokinetic parameters of AKH pure drug and proliposomal formulation.

PK Parameters	Pure Drug	Proliposomal Formulation (F8)
Tmax (h)	0.5	0.5
Cmax (ng/mL)	49.1	116.2
AUC_0-t_ (ng/mL/h)	166.8	382.7
Relative bioavailability (%)	-	230

## Data Availability

Not applicable.

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
