# Peer review of "Aliskiren Hemifumarate Proliposomes for Improved Oral Drug Delivery: Formulation Development, In Vitro and In Vivo Permeability Testing"

_molecules, 2022, doi:10.3390/molecules27154828_

Round 1

Reviewer 1 Report

- The manuscript contains yellow and green markings. Does that mean it wasn't the final version?

- Line 50: Please check this sentence

- Figure 1: Authors must correctly draw all structures. The current version is inadmisible.

- Authors are encouraged to include in the captions corresponding to tables and figures, to the extent possible, the methods used to achieve the results presented therein.

- Adding a list of abbreviations would be of great help to readers.

- Lines 314-316. Could the authors explain the effect of the mentioned electrostatic interactions? What is the charge on the aliphatic chain? Are they­­ referring to the charge of the amino group?

Reviewer 2 Report

The manuscript: molecules-1734976-peer-review-v1 can be accepted with minor revisions. Although the appropriate corrections requested by the previous reviewers have been made, there are further necessary changes:
- in fig. 2 the magnification is not indicated, and the image quality is low not allowing the focusing of the proliposomes, with the cryo TEM better images could be obtained.
- in fig. 4 the authors indicate the value of significance only in the legend of the figure, it is necessary to indicate, with an asterisk or other symbol, the points in which the significance is detected, not burdening the text with the description as in 2.6
- the same indication for figures 8 and 9, indicate the points of significance in the graph.
- all the figures presented by the authors are low resolution, this prevents the vision of the details of the X and Y axes
- there are still some typos in the text.

Author Response

Although the appropriate corrections requested by the previous reviewers have been made, there are further necessary changes:
Question: in fig. 2 the magnification is not indicated, and the image quality is low not allowing the focusing of the proliposomes, with the cryo TEM better images could be obtained.

Answer: Thank you for pointing out this. In the current version, we tried to take a better picture of the liposomes using our existing confocal microscope with 40X magnification. As we don’t have cryo TEM capability we included the best possible image in the current version. Request to Editor and reviewer: If the image quality is unacceptable, we are considering deleting the image and its figure number entirely from the text.

-Question: in fig. 4 the authors indicate the value of significance only in the legend of the figure, it is necessary to indicate, with an asterisk or other symbol, the points in which the significance is detected, not burdening the text with the description as in 2.6
- the same indication for figures 8 and 9, indicate the points of significance in the graph.
- all the figures presented by the authors are low resolution, this prevents the vision of the details of the X and Y axes

Answers: Thank you very much for bringing this up. This comment is beneficial in providing a better quality of data in the figures representing the significance.

In Figure 4, Figure 7 and Figure 8, the significance is represented in the graph using symbols (*, #, $). Also, under each figure we have mentioned the P values for significance and the matter related to significance had been deleted in the text. We also improved the resolution of the axes for the figures in the current version.

Question: there are still some typos in the text.

Answers: Verified the whole article and corrected the typos as we observed.

This manuscript is a resubmission of an earlier submission. The following is a list of the peer review reports and author responses from that submission.

Round 1

Reviewer 1 Report

Comments: Figure 2 needs to be improved for quality

Figure 4. release profile shows no difference between pure drug and F7 and F8 - need to perform anova analysis to show if there is any difference 

Need to show statistics for Figure 7 and 8. 

Section 5. conclusion is too brief needs to provide more details and overall outcome of the work.

Reviewer 2 Report

The authors explain that the experiments in the present paper are of higher quality than the experiments in the previous paper (Deshmukh DD, Ravis WR, Betageri GV. Improved delivery of cromolyn from oral proliposomal beads. Int J Pharm. 2008;358(1-2):128-136. doi:10.1016/j.ijpharm.2008.02.026.), but there is no mention of this fact in the present paper compared to the previous paper. I believe that the advantage of the current paper over the previous paper should be noted. I think it is unfair that there is no citation of the authors’ own paper in current version. If the authors claimed that proliposomes with different lipid composition and charge enhanced the effects in vivo and in vitro, they should discuss the differences more in discussion part. Especially, the authors selected new drug, AKH, as but both drug and c proliposomes with different lipid composition and charge changed, we do not understand if one of them will affect the permeability. Moreover, the authors consider the novel HPLC, LCMS, and a liquid-liquid extraction method are also distinct form previous manuscript, but this reviewer feel these results are not purpose of this article. If these results are the main purpose of this paper, the structure of the paper itself should be reconsidered.

Finally, I do not accept the current version of manuscript for publish to the Molecules.